# ESTIMATING TREATMENT EFFECTS IN NETWORKS US-ING DOMAIN ADVERSARIAL TRAINING

## ABSTRACT

Estimating heterogeneous treatment effects in network settings is complicated by interference, meaning that the outcome of an instance can be influenced by the treatment status of others. Existing causal machine learning approaches usually assume a known exposure mapping that summarizes how the outcome of a given instance is influenced by others' treatment, a simplification that is often unrealistic. Furthermore, the interaction between homophily—the tendency of similar instances to connect—and the treatment assignment mechanism can induce a network-level covariate shift that may lead to inaccurate treatment effect estimates, a phenomenon that has not yet been explicitly studied. To address these challenges, we propose HINet—a novel method that integrates graph neural networks with domain adversarial training. This combination allows estimating treatment effects under unknown exposure mappings while mitigating the impact of (network-level) covariate shift. An extensive empirical evaluation on synthetic and semi-synthetic network datasets demonstrates the effectiveness of our approach.

## 1 INTRODUCTION

Individualized treatment effect estimation enables data-driven optimization of decision-making in applications such as medicine (Feuerriegel et al., 2024), operations management (Vanderschueren et al., 2023), and economics (Varian, 2016). Traditionally, *no interference* is assumed, meaning that the treatment assigned to one instance does not affect the outcome of other instances (Imbens & Rubin, 2015; Rubin, 1980). However, this assumption is often violated in real-world settings due to *spillover effects* (Sobel, 2006; Forastiere et al., 2021), such as in vaccination, where a vaccine protects not only its recipient but also indirectly benefits their social contacts.

Recent advances in causal machine learning have introduced methods for estimating treatment effects in network settings (Ma & Tresp, 2021; Jiang & Sun, 2022; Chen et al., 2024). These methods often rely on a predefined exposure mapping, which specifies how the treatments of other instances in a network influence the outcome of a given instance (Aronow & Samii, 2017). A common choice is to define exposure as the sum or proportion of treated one-hop neighbors (Ma & Tresp, 2021; Forastiere et al., 2021; Jiang & Sun, 2022). While relying on such a predefined mapping simplifies the modeling of spillover effects, it is often unrealistic in real-world scenarios where the exact mechanisms behind these effects are unknown (Sävje, 2023). Moreover, spillover effects may be heterogeneous, i.e., dependent on the features of the instances involved (Huang et al., 2023; Zhao et al., 2024; Adhikari & Zheleva, 2025).

In this work, we propose *Heterogeneous Interference Network (HINet)*, a novel method that combines expressive GNN layers (Xu et al., 2019)—enabling the estimation of *heterogeneous spillover effects*—with domain adversarial training (Ganin et al., 2016; Bica et al., 2020) to *balance representations* for estimating treatment effects in the presence of interference, without relying on a predefined exposure mapping.

Additionally, we analyze how the interaction between homophily—the tendency of similar instances to connect—and the treatment assignment mechanism (e.g., a policy or self-selection) impacts the estimation of treatment effects in network settings with interference. This interaction can create clusters of treated and untreated nodes, introducing a *network-level covariate shift* (see Figure 1) in addition to the standard covariate shift between treated and control units. For example, older indi-

viduals may be more likely both to connect with each other in a social network (due to homophily) as well as to receive a vaccine (due to the treatment assignment mechanism).

**Contributions.** (1) We introduce HINet, a novel method for treatment effect estimation in network settings with interference. HINet combines expressive GNN layers—to learn an exposure mapping—with domain adversarial training—to address (network-level) covariate shift; (2) we empirically demonstrate HINet's effectiveness in estimating treatment effects in the presence of interference with unknown exposure mappings; (3) we propose two new metrics for evaluating treatment effect estimates in these scenarios; and (4) we empirically show that domain adversarial training reduces the impact of the covariate shift resulting from the interaction between homophily and a treatment assignment mechanism on treatment effect estimation.

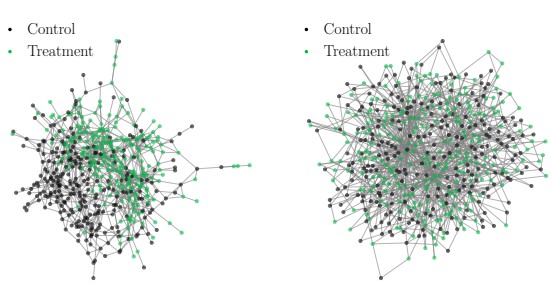

(a) Network w/ homophily    (b) Network w/o homophily

Figure 1: Homophily and the treatment assignment mechanism interact to create clusters of treated and untreated nodes within the network, i.e., network-level covariate shift. When there is no homophily, the treatment is allocated randomly with respect to network topology.

## 2 RELATED WORK

**Heterogeneous treatment effect estimation.** A large body of work has been dedicated to Conditional Average Treatment Effect (CATE) estimation (Hill, 2011; Shalit et al., 2017; Künzel et al., 2019; Curth & Van der Schaar, 2021; Vanderschueren et al., 2025). In contrast to the Average Treatment Effect (ATE), CATEs capture heterogeneity across subpopulations, enabling tailored treatment allocation decisions (Feuerriegel et al., 2024). A key challenge in estimating heterogeneous effects from observational data is the covariate shift between treated and control units induced by the treatment assignment mechanism, which can lead to inaccurate treatment effect estimates (Johansson et al., 2016; 2022; Shalit et al., 2017). To address this challenge, various machine learning methods have been developed, such as propensity weighting and balancing the representations of the treatment and control groups (Shalit et al., 2017; Yao et al., 2018; Hassanpour & Greiner, 2019). One notable method closely related to our work is CRN (Bica et al., 2020), which employs an adversarial representation balancing approach for estimating treatment effects over time.

**Heterogeneous treatment effect estimation in the presence of interference.** Recently, several methods have been developed to estimate treatment effects in network settings with interference. These methods often rely on a predefined exposure mapping that uses a basic aggregation function to summarize the treatments of the neighbors of an instance into a single variable that affects the outcome of the instance (Aronow & Samii, 2017). Most methods use the proportion of treated one-hop neighbors as mapping function. Under this assumption, a variety of estimators have been proposed, including inverse probability weighted (IPW) (Forastiere et al., 2021) and doubly robust (Chen et al., 2024) estimation. Other methods use graph neural networks (GNNs) and balance the representations of the treatment and control groups by incorporating an Integral Probability Metric (IPM) into the loss function (Ma & Tresp, 2021; Cai et al., 2023), while Jiang & Sun (2022) employs adversarial training.

Since the assumption of a predefined exposure mapping may be unrealistic, there has been growing interest in *learning* this mapping directly from data. IDE-Net (Adhikari & Zheleva, 2025) uses multiple plausible and expressive candidate mappings and concatenates them, allowing the model to learn a weighted combination that best approximates the true exposure mapping. SPNet (Huang et al., 2023) leverages masked attention, while Zhao et al. (2024) combine attention weights with Dual Weighted Regression to address covariate shift. In contrast, we propose an alternative approach that integrates Graph Isomorphism Networks (Xu et al., 2019) with domain adversarial training (Ganin et al., 2016; Bica et al., 2020) to mitigate the impact of (network-level) covariate shift.

A related line of research examines treatment effects in the presence of interference over time, i.e., contagion effects, where the outcomes of different entities can also influence each other over time (Jiang et al., 2023; Fatemi & Zheleva, 2023). Another strand of work, which maintains the no-interference assumption, leverages network information to mitigate confounding bias in CATE estimation (Guo et al., 2020; 2021).

**Homophily in causal inference.** Disentangling causal (network) effects from homophily is a well-known and important problem in network influence research (Shalizi & Thomas, 2011; Aral et al., 2009; McFowland III & Shalizi, 2023). The problem is that, when latent homophily is present, it becomes impossible to identify whether an observed outcome is caused by network influence or by latent homophily. For example, obese individuals have been shown to cluster in social networks (Christakis & Fowler, 2007). However, identifying whether this pattern arises from latent homophily or from actual contagion of obesity is very difficult using observational data (Shalizi & Thomas, 2011; Ogburn et al., 2024).

## 3 PROBLEM SETUP

**Notation.** We consider an undirected network $\mathcal{G} = \{\mathcal{V}, \mathcal{E}\}$ where $\mathcal{V}$ is the set of nodes and $\mathcal{E}$ the set of edges connecting the nodes. The set of edges of node $i$ is denoted as $\mathcal{E}_i$. Each node $i$ is an instance or unit in the network with covariates $\mathbf{X}_i \in \mathcal{X} \subseteq \mathbb{R}^d$, a treatment $T_i \in \mathcal{T} = \{0, 1\}$, and an outcome $Y_i \in \mathcal{Y} \subseteq \mathbb{R}$. In marketing, for example, $\mathbf{X}_i$ can represent customer features, $T_i$ whether a customer was targeted with a marketing campaign, and $Y_i$ customer expenditure. The set of directly connected instances, or one-hop neighbors, of instance $i$ are denoted $\mathcal{N}_i$. $\mathcal{N}_i$ is used as a subscript to describe the set of covariates $\mathbf{X}_{\mathcal{N}_i} = \{\mathbf{X}_j\}_{j \in \mathcal{N}_i}$ or treatments $\mathbf{T}_{\mathcal{N}_i} = \{T_j\}_{j \in \mathcal{N}_i}$ of $i$'s neighbors. The potential outcome for unit $i$ with treatment $t_i$ and the set of treatments of its neighbors $\mathbf{t}_{\mathcal{N}_i}$ is denoted as $Y_i(t_i, \mathbf{t}_{\mathcal{N}_i})$.

**Assumptions.** We adopt the Markov assumption: only directly connected instances influence each other. The Directed Acyclic Graph (DAG) of the assumed causal structure is visualized in Figure 2 (Greenland et al., 1999; Ogburn & VanderWeele, 2014). Three mutually connected instances $i, j$, and $k$ are shown. The features of a unit $i$, $\mathbf{X}_i$, influence both the treatment $T$ and outcome $Y$ of itself as well as of its neighbors. The treatment $T_i$, in turn, affects the outcome of both itself and its neighbors. The arrows from $\mathbf{X}_k$ and $T_k$ to $Y_j$, and from $\mathbf{X}_j$ and $T_j$ to $Y_k$ are omitted for visual clarity.

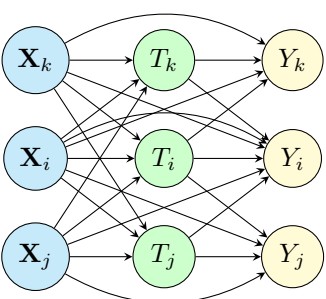

Figure 2: DAG of the assumed causal structure.

We further assume access to observational data $\mathcal{D} = \left( \{\mathbf{x}_i, t_i, y_i, \}_{i=1}^{|\mathcal{V}|}; \mathcal{G} \right)$. Importantly, this data does not necessarily come from a randomized controlled trial (RCT), and a treatment assignment mechanism might be present, as represented in the DAG in Figure 2 by the arrows from a unit's features to its own treatment and the treatments of its neighbors.

For the exposure mapping, i.e., a function $z : \mathcal{T}^{\mathcal{N}_i} \times \mathcal{X}^{\mathcal{N}_i} \to \mathbb{R}$ that maps treatments of neighbors in the network, together with their relevant features, to an exposure $z_i$ (Aronow & Samii, 2017), we only assume its existence. Hence, in contrast to other recent work (Ma & Tresp, 2021; Jiang & Sun, 2022; Chen et al., 2024) that assumes a predefined form, we estimate this function from data.

Finally, the classical assumptions from causal inference are slightly modified to ensure identifiability in a network setting (Forastiere et al., 2021; Jiang & Sun, 2022):

*Consistency:* If $T_i = t_i$ and $\mathbf{T}_{\mathcal{N}_i} = \mathbf{t}_{\mathcal{N}_i}$, then $Y_i = Y_i(t_i, \mathbf{t}_{\mathcal{N}_i})$.

*Overlap:* $\exists \delta \in (0, 1)$ such that $\delta < p(T_i = t_i, \mathbf{T}_{\mathcal{N}_i} = \mathbf{t}_{\mathcal{N}_i} \mid \mathbf{X}_i = \mathbf{x}_i, \mathbf{X}_{\mathcal{N}_i} = \mathbf{x}_{\mathcal{N}_i}) < 1 - \delta$.

*Strong ignorability:* $Y_i(t_i, \mathbf{t}_{\mathcal{N}_i}) \perp\!\!\!\perp T_i, \mathbf{T}_{\mathcal{N}_i} \mid \mathbf{X}_i, \mathbf{X}_{\mathcal{N}_i}, \forall t_i \in \mathcal{T}, \mathbf{t}_{\mathcal{N}_i} \in \mathcal{T}^{\mathcal{N}_i}, \mathbf{X}_i \in \mathcal{X}, \mathbf{X}_{\mathcal{N}_i} \in \mathcal{X}^{\mathcal{N}_i}$.

**Objective.** We aim to estimate the Individual Total Treatment Effect (ITTE) (Caljon et al., 2025), defined as:

$$\omega_i(t_i, \mathbf{t}_{\mathcal{N}_i}) = \mathbb{E}\big[Y_i(t_i, \mathbf{t}_{\mathcal{N}_i}) - Y_i(0, \mathbf{0}) \mid \mathbf{x}_i, \mathbf{x}_{\mathcal{N}_i}\big]. \tag{1}$$

To this end, we train a model $\mathcal{M}(\mathbf{x}_i, t_i, \mathbf{x}_{\mathcal{N}_i}, \mathbf{t}_{\mathcal{N}_i})$ to predict $Y_i(t_i, \mathbf{t}_{\mathcal{N}_i})$, which is plugged into the defintion of ITTE to obtain $\hat{\omega}_i(t_i, \mathbf{t}_{\mathcal{N}_i}) = \mathcal{M}(\mathbf{x}_i, t_i, \mathbf{x}_{\mathcal{N}_i}, \mathbf{t}_{\mathcal{N}_i}) - \mathcal{M}(\mathbf{x}_i, 0, \mathbf{x}_{\mathcal{N}_i}, \mathbf{0})$.

**Interaction between homophily and the treatment assignment mechanism.** In observational data, the treatment assignment mechanism—such as a policy or self-selection—can induce *covariate shift*, meaning that the treatment and control groups have different covariate distributions. As a result, an additional source of variance of treatment effect estimates is introduced (Johansson et al., 2016; Shalit et al., 2017). In a network setting with interference, this issue can also arise and may even be amplified by homophily. Homophily is a social phenomenon referring to the tendency of individuals with similar features to be connected in a social network (McPherson et al., 2001). When the features that drive homophily also influence treatment assignment, an additional form of covariate shift emerges—namely, *network-level covariate shift*. This occurs because similar instances are not only more likely to be connected but also more likely to receive the same treatment. Consequently, an instance that is likely to be treated (due to the treatment assignment mechanism) is also more likely to have treated neighbors (due to the interaction between homophily and the treatment assignment mechanism). This can create clusters of treated and untreated instances within a network, as depicted in Figure 1a, resulting in network-level covariate shift, where instances with many treated neighbors and those with few treated neighbors have different feature distributions. We hypothesize that this can lead to more inaccurate treatment effect estimates and that learning balanced node representations can help reduce estimation error. We empirically investigate this in Section 5.2.

For homophilous networks, the DAG in Figure 2 changes. Connections between individuals are no longer exogenous but instead dependent on their characteristics through homophily. This means that conditioning on a node's connections creates a collider structure, inducing an association between the features of those nodes (Pearl, 2009). Nevertheless, under the strong ignorability assumption, treatment effects remain identifiable. More details are provided in Appendix B.

## 4 METHODOLOGY

**Architecture.** The *architecture of HINet*—our proposed neural method for estimating treatment effects in the presence of interference, which models heterogeneous spillover effects and uses domain adversarial training to learn balanced representations—is visualized in Figure 3. Following Bica et al. (2020) and Berrevoets et al. (2020), the neural network first learns node representations and then splits into two branches. More specifically, for each node $k \in \{i\} \cup \mathcal{N}_i$, the features $\mathbf{x}_k$ are transformed into a representation $\phi_k = e_\phi(\mathbf{x}_k)$ via a multi-layer perceptron (MLP), and these representations are then used to predict both the treatment $\hat{t}_i$ and the outcome $\hat{y}_i$ of node $i$.

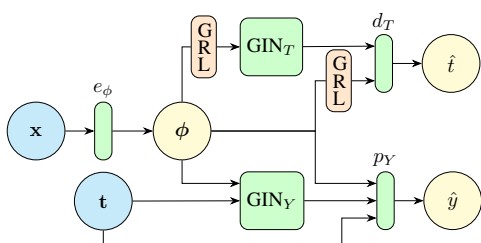

Figure 3: HINet architecture.

The *lower branch* predicts the outcome of node $i$. To learn the exposure mapping and to account for other network information, a Graph Isomorphism Network (GIN) is used (Xu et al., 2019). This module takes as input the node representations $\phi_k$ and treatments $t_k$ for $k \in \{i\} \cup \mathcal{N}_i$. Unlike some other GNN architectures, such as GCN (Kipf & Welling, 2016) and GraphSAGE (Hamilton et al., 2017), GIN offers maximal representational capacity, making it particularly well-suited for learning different exposure mapping functions and heterogeneous spillover effects. $\text{GIN}_Y$ first concatenates the node representations and treatments for all nodes $k$, i.e., $c_k = \phi_k \oplus t_k, \ \forall k$, and then outputs

$$\text{MLP}\big((1 + \epsilon) \cdot c_i + \sum_{j \in \mathcal{N}_i} c_j\big). \tag{2}$$

This $\text{GIN}_Y$ output is subsequently combined with $\phi_i$ and $t_i$ (for the direct treatment effect), and fed into the MLP $p_Y$ to generate the outcome prediction $\hat{y}_i$.

The *upper branch* is used to learn treatment-invariant, or balanced, representations. Its output is the predicted treatment $\hat{t}_i$. Its setup is similar to that of the lower branch, with two key differences: treatments are not used as inputs, and Gradient Reversal Layers (GRLs) (Ganin et al., 2016) are used. GRLs do not affect the forward pass but reverse the gradient in the backward pass.

**Loss function.** HINet is trained by combining two different losses: the outcome loss and the treatment prediction loss, defined respectively as $\mathcal{L}_y = \frac{1}{n} \sum_{i=1}^n (y_i - \hat{y}_i)^2$ and $\mathcal{L}_t = \frac{1}{n} \sum_{i=1}^n \text{BCE}(t_i, \hat{t}_i)$, where BCE is the binary cross-entropy loss. Thanks to the GRL, we can optimize the combined loss

$$\mathcal{L}_{\text{comb}} = \mathcal{L}_y + \alpha \cdot \mathcal{L}_t, \tag{3}$$

where $\alpha$ determines the importance of adversarial balancing. Note that $\mathcal{L}_y$ does not affect the upper (treatment) branch, while $\mathcal{L}_t$ does not affect the lower (outcome) branch. However, both affect $e_\phi$.

**Representation balancing.** More formally, HINet aims to learn node representations $\phi_i$ that are invariant to their treatments:

$$p(\phi_i, \phi_{\mathcal{N}_i} \mid T_i = t_i) = p(\phi_i, \phi_{\mathcal{N}_i}) \quad \forall i \in \mathcal{V}. \tag{4}$$

Given our assumptions on the causal graph and Equation (4), invariance to the treatments of each of its neighbors is also induced (see Appendix A for proof):

$$p(\phi_i, \phi_{\mathcal{N}_i} \mid T_j = t_j) = p(\phi_i, \phi_{\mathcal{N}_i}) \quad \forall i \in \mathcal{V}, j \in \mathcal{N}_i. \tag{5}$$

Together, these properties imply that the learned representations are invariant not only to a node's own treatment but also to the treatments of its neighbors. This yields node representations that are effectively treatment-invariant with respect to the marginal distributions of all treatments, which in turn aids in reducing the impact of network-level covariate shift on estimation accuracy (Shalit et al., 2017; Bica et al., 2020; Berrevoets et al., 2020). We empirically validate this effect in Section 5.4. Nevertheless, because of the trade-off in the loss function between predictive accuracy and treatment invariance, the resulting representations are not guaranteed to be perfectly treatment-invariant.

**Measuring model performance in the presence of interference.** In a traditional no-interference setting with binary treatment, the Precision in Estimation of Heterogeneous Effects (PEHE) (Hill, 2011) is often used to evaluate treatment effect estimates on (semi-)synthetic data. PEHE is defined as the root mean squared error of the estimated CATEs, which is uniquely defined since there is a single counterfactual. In a network setting, this is no longer the case as there are *many possible treatment assignments*, each resulting in a different potential outcome. How to best evaluate treatment effect estimation methods in the presence of interference using (semi-)synthetic data remains an open question. Previous work has typically assessed performance based on only one *counterfactual network*—i.e., a network in which at least one node receives a different treatment than in the observed network (see, e.g., Jiang & Sun (2022); Chen et al. (2024)). However, some models may be accurate for certain counterfactual networks (e.g., those with a low treatment rate) but perform poorly for others. Therefore, we argue that a good evaluation procedure should account for *performance across multiple counterfactual networks*. Yet, since there are $2^{|\mathcal{V}|} - 1$ possible counterfactual networks, it is computationally infeasible to evaluate all of these for large networks.

To address this issue, we propose two *novel evaluation metrics* inspired by the Mean Integrated Squared Error (MISE) (Schwab et al., 2020), which is used for evaluating continuous treatment effect estimates: the *Precision in Estimation of Heterogeneous Network Effects (PEHNE)* and the *Counterfactual Network Estimation Error (CNEE)*. The first measures estimation error in terms of ITTE, while the second uses counterfactual outcomes. For both metrics, we sample $m$ counterfactual networks and calculate the estimation error for each node within each sampled counterfactual network. The final score is obtained by averaging across all nodes and all sampled counterfactual networks. The key difference is that CNEE places less emphasis on the estimation of potential outcomes without any treatment, $Y_i(0, \mathbf{0})$. Further details, including pseudocode, are provided in Appendix C.

## 5 EXPERIMENTS AND DISCUSSION

**Data.** Synthetic and semi-synthetic data are commonly used in causal machine learning to evaluate treatment effect estimators, as ground truth effects are unobservable in real-world datasets (Berrevoets et al., 2020; Feuerriegel et al., 2024). Following related work (Ma & Tresp, 2021;

Jiang & Sun, 2022; Chen et al., 2024), we use the *Flickr* and *BlogCatalog (BC)* datasets for semi-synthetic dataset generation. To further evaluate generalization across different network structures, we also simulate two fully synthetic datasets: one using the *Barabási–Albert (BA Sim)* random network model (Barabási & Albert, 1999), and another using a procedure that generates *homophilous (Homophily Sim)* graphs based on cosine similarity. The latter allows us to examine the potential impact of homophily-induced network-level covariate shift on estimation accuracy.

For each dataset, a training, validation, and test set is generated. Each fully synthetic dataset (BA Sim and Homophily Sim) contains 5,000 nodes per split, while the real-world datasets have smaller splits: Flickr contains approximately 2,400 nodes per split, and BC contains about 1,700 nodes per split. For the first two experiments, a weighted sum depending on the node features is used as the exposure mapping, i.e., $z_i = \frac{1}{|\mathcal{N}_i|} \sum_{j \in \mathcal{N}_i} w(\mathbf{x}_j) t_j$, with $w(\cdot)$ a function that maps the features of each node to a weight. Note, however, that HINet makes no assumptions about the form of the exposure mapping. Full details on the data-generating processes (DGPs) are provided in Appendix D.

To assess whether network-level covariate shift is present in our datasets, we measure treatment and outcome assortativity (Newman, 2002; 2003). Intuitively, treatment (outcome) assortativity quantifies whether nodes with similar treatments (outcomes) are more likely to be connected in a network than would be expected by chance, yielding a score between -1 and 1. Table 1 shows that in the Homophily Sim dataset, both treatment and outcome assortativity are positive, indicating that nodes with similar treatments or outcomes are more likely to be connected and that a network-level covariate shift is present. In the other datasets, assortativities are close to zero. Appendix E provides more details on the quantification of homophily.

|                        | BC   | Flickr | BA Sim | Homophily Sim |
|------------------------|------|--------|--------|---------------|
| Treatment assortativity | 0.06 | 0.00   | -0.01  | 0.56          |
| Outcome assortativity   | 0.06 | -0.01  | -0.17  | 0.71          |

Table 1: Treatment and outcome assortativity for the different datasets used in our experiments.

**Methods for comparison.** We compare HINet to the following methods for estimating treatment effects: *TARNet* (Shalit et al., 2017), which ignores network information; *NetDeconf* (Guo et al., 2020), which incorporates network information but does not account for spillover effects; *NetEst* (Jiang & Sun, 2022), which relies on a predefined exposure mapping to estimate spillover effects; *TNet* (Chen et al., 2024), which also relies on a predefined exposure mapping, but leverages targeted learning for doubly robust estimation of spillover effects; and *SPNet* (own implementation) (Zhao et al., 2024), which estimates heterogeneous spillover effects using a masked attention mechanism. Finally, we also include a *GIN model*, which uses node features and treatments as inputs to a GIN layer that is followed by an MLP that outputs a prediction $\hat{y}_i$. This baseline is included to contrast our method with vanilla graph machine learning methods. The main difference with HINet is that it does not use balancing. Following the literature, we use $z_i^{\text{assumed}} = \frac{1}{|\mathcal{N}_i|} \sum_{j \in \mathcal{N}_i} t_j$ as the assumed exposure mapping in TNet and NetEst. Given that the true exposure mapping is unknown, these two methods use a misspecified exposure mapping for most experiments.

**Hyperparameter selection.** Selecting hyperparameters in causal machine learning is challenging due to the fundamental problem of causal inference: treatment effects cannot be directly observed, making it impossible to optimize models for treatment effect estimation accuracy. As a result, metrics such as PEHNE and CNEE cannot be used during model selection. Instead, alternative metrics must be used for hyperparameter selection (Curth & van der Schaar, 2023; Vanderschueren et al., 2025). In this work, we tune all hyperparameters—except the treatment prediction loss weight $\alpha$—using the factual validation loss, i.e., the prediction error on the observed outcomes in the validation set. Selecting $\alpha$ via the factual loss would typically lead to $\alpha = 0$, since a positive $\alpha$ may cause the model to discard relevant information for predicting the observed outcomes in favor of constructing treatment-invariant representations. However, both theoretical and empirical work indicate that representation balancing can improve treatment effect estimation (Shalit et al., 2017; Bica et al., 2020; Berrevoets et al., 2020). Therefore, we determine $\alpha$ heuristically by selecting the largest value for which the factual validation loss is below $(1 + p) \cdot \text{loss}_{\alpha=0}$. As a rule of thumb, we set $p = 0.10$, allowing for a maximum increase in validation error of 10% compared to setting $\alpha = 0$. An important advantage of this approach is that it allows for $\alpha = 0$ when representation balancing would result in excessive information loss. More details are provided in Appendix F.

## 5.1 PERFORMANCE ON (SEMI-)SYNTHETIC DATA

Table 2 reports the test set results for all datasets in terms of the PEHNE and CNEE metrics. HINet achieves the lowest error in terms of both metrics on all datasets except BA Sim, demonstrating its superior ability to estimate treatment effects in the presence of interference under an unknown exposure mapping. On the BA Sim dataset, TNet performs best in terms of PEHNE, but not in terms of CNEE, suggesting that TNet is better at predicting $Y_i(0, \mathbf{0})$ than HINet for this dataset. A possible explanation is that TNet assumes the correct exposure mapping when no neighbors are treated (i.e., $z_i = 0$). Nevertheless, since HINet still attains the lowest CNEE, it is on average more accurate at predicting potential outcomes across counterfactual networks than TNet, which relies on a misspecified exposure mapping.

| Dataset | Metric | TARNet | NetDeconf | NetEst | TNet | GIN model | SPNet | HINet (ours) |
|---|---|---|---|---|---|---|---|---|
| BC | PEHNE | $3.45 \pm 0.03$ | $5.52 \pm 0.24$ | $2.76 \pm 0.23$ | $1.95 \pm 0.20$ | $\underline{1.33 \pm 0.18}$ | $4.93 \pm 0.37$ | $\mathbf{0.77 \pm 0.18}$ |
| | CNEE | $4.13 \pm 0.04$ | $5.53 \pm 0.23$ | $2.82 \pm 0.23$ | $2.28 \pm 0.26$ | $\underline{1.20 \pm 0.17}$ | $5.11 \pm 0.46$ | $\mathbf{0.80 \pm 0.20}$ |
| Flickr | PEHNE | $3.85 \pm 0.10$ | $5.67 \pm 0.09$ | $2.67 \pm 0.16$ | $1.30 \pm 0.16$ | $\underline{1.07 \pm 0.09}$ | $6.35 \pm 0.22$ | $\mathbf{0.65 \pm 0.06}$ |
| | CNEE | $6.17 \pm 0.20$ | $7.21 \pm 0.14$ | $4.20 \pm 0.21$ | $2.07 \pm 0.04$ | $\underline{0.99 \pm 0.09}$ | $7.76 \pm 0.13$ | $\mathbf{0.73 \pm 0.09}$ |
| BA Sim | PEHNE | $3.03 \pm 0.02$ | $4.61 \pm 0.03$ | $0.85 \pm 0.08$ | $\mathbf{0.68 \pm 0.03}$ | $1.39 \pm 0.04$ | $4.11 \pm 0.05$ | $\underline{0.80 \pm 0.08}$ |
| | CNEE | $5.96 \pm 0.01$ | $6.66 \pm 0.03$ | $1.79 \pm 0.11$ | $1.26 \pm 0.02$ | $\underline{1.22 \pm 0.04}$ | $6.02 \pm 0.04$ | $\mathbf{0.85 \pm 0.08}$ |
| Homophily Sim | PEHNE | $2.23 \pm 0.01$ | $1.09 \pm 0.05$ | $\underline{0.62 \pm 0.09}$ | $0.93 \pm 0.06$ | $0.89 \pm 0.09$ | $1.75 \pm 0.09$ | $\mathbf{0.22 \pm 0.02}$ |
| | CNEE | $4.49 \pm 0.01$ | $1.21 \pm 0.05$ | $\underline{0.86 \pm 0.10}$ | $0.95 \pm 0.04$ | $1.06 \pm 0.09$ | $1.73 \pm 0.09$ | $\mathbf{0.23 \pm 0.02}$ |

Table 2: Test set results (mean $\pm$ SD over five different initializations). Lower is better for both metrics. The best-performing method is in bold; the second-best is underlined.

## 5.2 IMPACT OF HOMOPHILY

Figure 4 presents the performance of HINet on the BA Sim (No Homophily) and Homophily Sim (Homophily) networks across three different DGPs and varying levels of treatment assignment mechanism strength $\beta_{XT}$ ($\beta_{XT} = 0$ resembles an RCT). The DGPs are: (a) individual (direct) treatment effects without interference, (b) spillover effects without direct effects, and (c) both individual and spillover effects. CNEE test set results are visualized with and without ($\alpha = 0$) balancing node representations. As expected, CNEE generally increases with rising $\beta_{XT}$, due to stronger covariate shift (Shalit et al., 2017). Notably, we also observe that representation balancing has a greater positive impact on performance in the homophilous networks compared to the non-homophilous ones when spillover effects—both with and without individual effects—are present. These findings support our hypothesis that balancing node representations can partly offset the estimation error introduced by network-level covariate shift. The results in terms of PEHNE, provided in Appendix G.1, are similar.

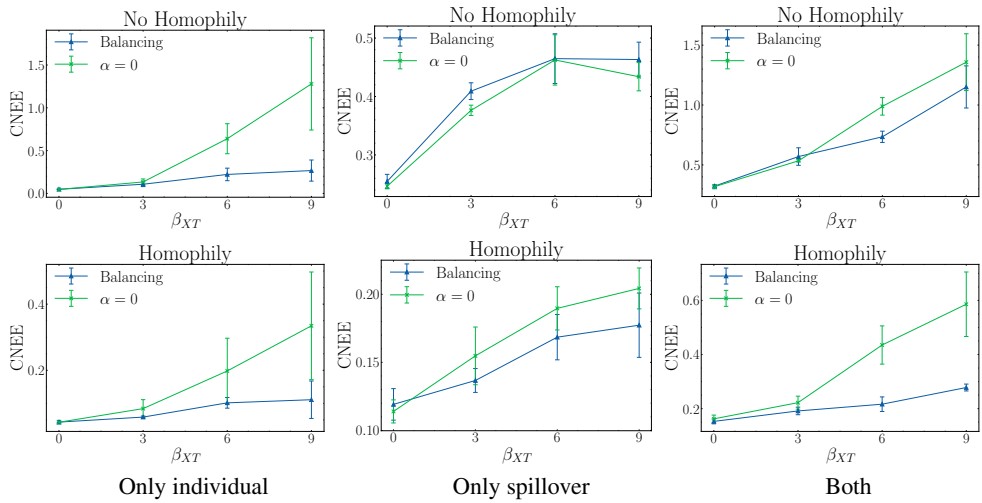

Figure 4: Impact of balancing node representations on test CNEE (mean $\pm$ SD over five different initializations). The two rows correspond to the BA and Homophily Sim datasets. The columns correspond to different DGPs. The x-axis shows increasing treatment assignment mechanism strength $\beta_{XT}$. Lower values indicate better performance.

## 5.3 LEARNING EXPOSURE MAPPINGS

A key advantage of HINet is its ability to learn an exposure mapping without imposing assumptions on its functional form. To substantiate this claim, we replicate the experiment from Section 5.1 using alternative exposure mappings in the DGP.

Table 3 reports results for the exposure mapping $z_i = \sum_{j \in \mathcal{N}_i} t_j$. The results show that both NetEst and TNet perform poorly because their assumed exposure mappings differ substantially from the true one. This highlights how misspecification of the exposure mapping can lead to inaccurate treatment effect estimates. In contrast, HINet and the GIN model—both of which do not rely on assumptions regarding the exposure mapping—achieve the best performance.

| Dataset | Metric | TARNet | NetDeconf | NetEst | TNet | GIN model | SPNet | HINet (ours) |
|---|---|---|---|---|---|---|---|---|
| BC | PEHNE | 572.78 ± 12.24 | 625.37 ± 15.01 | 324.58 ± 29.46 | 541.69 ± 70.55 | 51.31 ± 8.46 | 552.80 ± 4.83 | **25.22 ± 1.69** |
|  | CNEE | 325.18 ± 4.22 | 408.75 ± 9.23 | 261.20 ± 12.54 | 507.97 ± 82.85 | 40.43 ± 5.87 | 370.72 ± 17.83 | **23.57 ± 1.28** |
| Flickr | PEHNE | 2854.51 ± 8.34 | 2865.66 ± 77.69 | 2817.39 ± 79.31 | $2.25 \cdot 10^9 \pm 3.65 \cdot 10^9$ | 306.11 ± 34.20 | 2736.06 ± 12.54 | **229.72 ± 37.89** |
|  | CNEE | 2540.88 ± 16.33 | 2634.20 ± 59.32 | 2505.10 ± 21.09 | $2.12 \cdot 10^9 \pm 3.38 \cdot 10^9$ | 271.94 ± 28.31 | 1013.38 ± 59.93 | **231.86 ± 32.76** |
| BA Sim | PEHNE | 71.60 ± 0.32 | 76.05 ± 0.12 | 55.05 ± 0.30 | 64.07 ± 10.85 | **10.75 ± 0.62** | 76.24 ± 0.89 | 11.61 ± 0.69 |
|  | CNEE | 61.33 ± 0.11 | 64.53 ± 0.24 | 53.76 ± 0.28 | 59.38 ± 8.60 | **9.93 ± 0.47** | 38.23 ± 0.95 | 12.20 ± 0.86 |
| Homophily Sim | PEHNE | 32.32 ± 0.16 | 35.94 ± 1.30 | 7.19 ± 0.30 | 9.82 ± 0.53 | 0.63 ± 0.15 | 23.61 ± 0.35 | **0.20 ± 0.07** |
|  | CNEE | 18.89 ± 0.09 | 27.63 ± 1.34 | 6.73 ± 0.25 | 9.38 ± 0.43 | 0.61 ± 0.12 | 17.02 ± 0.26 | **0.22 ± 0.07** |

Table 3: Test set results (mean ± SD over five different initializations) for the **sum** of treatments of neighbors used as exposure mapping in the DGP. Lower is better for both metrics. The best-performing method is in bold; the second-best is underlined.

Table 4 reports results for the exposure mapping $z_i = \frac{1}{|\mathcal{N}_i|} \sum_{j \in \mathcal{N}_i} t_j$, which is the exposure mapping assumed by TNet and NetEst. For this DGP, the performance of both methods is much closer to that of HINet. Interestingly, however, neither method matches nor surpasses HINet's performance, despite relying on a correctly specified exposure mapping. A possible explanation is that TNet and NetEst are unable to capture the nonlinear nature of the effect of $\mathbf{X}_{\mathcal{N}_i}$ on $Y_i$, which is present in the DGP of this experiment (see Appendix D). To substantiate this, we perform another experiment in which we set the influence of $\mathbf{X}_{\mathcal{N}_i}$ on $Y_i$ to zero. The results for this simplified DGP, presented in Appendix G.2.1, show that TNet indeed becomes the best-performing model in this setting.

| Dataset | Metric | TARNet | NetDeconf | NetEst | TNet | GIN model | SPNet | HINet (ours) |
|---|---|---|---|---|---|---|---|---|
| BC | PEHNE | 4.10 ± 0.03 | 5.39 ± 0.19 | 1.48 ± 0.16 | 1.50 ± 0.28 | 1.42 ± 0.23 | 4.94 ± 0.28 | **0.62 ± 0.20** |
|  | CNEE | 3.77 ± 0.03 | 5.39 ± 0.19 | 1.44 ± 0.16 | 1.72 ± 0.32 | 1.14 ± 0.20 | 4.90 ± 0.36 | **0.59 ± 0.15** |
| Flickr | PEHNE | 4.04 ± 0.02 | 4.30 ± 0.06 | 2.28 ± 0.18 | 1.15 ± 0.23 | 1.27 ± 0.10 | 5.02 ± 0.24 | **0.50 ± 0.16** |
|  | CNEE | 4.80 ± 0.01 | 5.63 ± 0.09 | 3.69 ± 0.22 | 1.46 ± 0.09 | 1.14 ± 0.12 | 6.29 ± 0.16 | **0.60 ± 0.17** |
| BA Sim | PEHNE | 3.28 ± 0.02 | 3.60 ± 0.03 | 0.41 ± 0.08 | **0.12 ± 0.02** | 1.38 ± 0.06 | 3.29 ± 0.03 | 0.21 ± 0.05 |
|  | CNEE | 4.53 ± 0.01 | 5.02 ± 0.04 | 1.10 ± 0.11 | 0.56 ± 0.03 | 1.12 ± 0.08 | 4.56 ± 0.02 | **0.23 ± 0.06** |
| Homophily Sim | PEHNE | 3.56 ± 0.04 | 1.58 ± 0.07 | 0.36 ± 0.02 | 0.09 ± 0.02 | 0.50 ± 0.08 | 1.87 ± 0.14 | **0.07 ± 0.01** |
|  | CNEE | 3.49 ± 0.02 | 1.51 ± 0.07 | 0.54 ± 0.03 | 0.22 ± 0.02 | 0.60 ± 0.08 | 1.71 ± 0.14 | **0.09 ± 0.01** |

Table 4: Test set results (mean ± SD over five different initializations) for the **proportion** of treated neighbors used as exposure mapping in the DGP. Lower is better for both metrics. The best-performing method is in bold; the second-best is underlined.

Additional results for several other alternative exposure mappings are provided in Appendix G.2.2 and further demonstrate the superior performance of HINet.

## 5.4 ABLATION STUDY

**Representation balancing.** To assess the impact of balancing node representations on treatment effect estimation error, we report results from an experiment in which specific components of the upper (treatment) branch of HINet (see Figure 3) are removed. Three variants of HINet are compared using the same datasets as in Section 5.1, with the same exposure mapping but increasing treatment assignment mechanism strength $\beta_{XT}$. We compare: (1) HINet with all its components, (2) HINet without balancing ($\alpha = 0$), and (3) HINet without $\text{GIN}_T$ (No $\text{GIN}_T$), and hence only balances the individual node representations with respect to their own treatment. Therefore, in this last variant, Equation (5) no longer holds, and the representations are not balanced with respect to the treatments of neighbors.

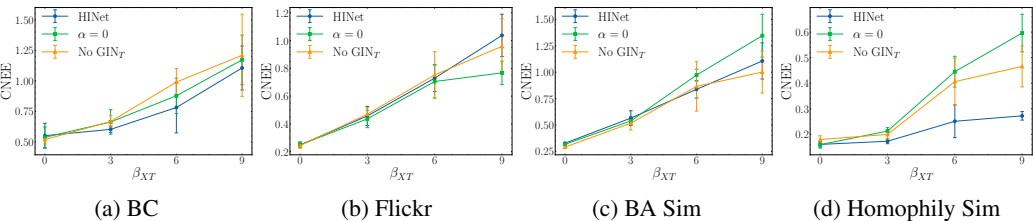

| (a) BC | (b) Flickr | (c) BA Sim | (d) Homophily Sim |

Figure 5: CNEE across datasets with increasing treatment assignment mechanism strength $\beta_{XT}$.

The results in Figure 5 indicate that balancing improves treatment effect estimation accuracy for the two larger datasets, BA Sim and Homophily Sim. For the considerably smaller semi-synthetic datasets, balancing has little effect, with even a slightly negative effect observed for the Flickr dataset. Nevertheless, for larger datasets—and especially when homophily is present—balancing improves estimation accuracy. Notably, balancing with respect to neighbors' treatments (via $\text{GIN}_T$) appears to matter most when homophily—and thus network-level covariate shift—is present, as evidenced by the results for Homophily Sim. These findings provide additional evidence that this type of balancing helps mitigate the estimation error caused by network-level covariate shift.

**GNN architectures.** Prior work on treatment effect estimation in network settings has primarily relied on Graph Convolutional Networks (GCNs) (Jiang & Sun, 2022; Chen et al., 2024), which have limited representational capacity. Other GNN architectures, such as Graph Attention Networks (GATs) (Velickovic et al., 2018) and GraphSAGE (Hamilton et al., 2017), can learn more flexible aggregation functions, as is the case for GINs. To assess the impact of using GINs, we report results from an experiment with the same experimental settings as in Section 5.1, in which the GIN components of HINet, $\text{GIN}_T$ and $\text{GIN}_Y$, are replaced with these alternative architectures.

| Dataset | Metric | GIN | GAT | GraphSAGE | GCN |
|---|---|---|---|---|---|
| BC | PEHNE | **0.77 ± 0.18** | 1.20 ±0.39 | 0.82±0.19 | 0.79±0.14 |
| | CNEE | 0.80 ± 0.20 | 1.21 ±0.33 | **0.76±0.17** | 0.94±0.19 |
| Flickr | PEHNE | 0.65 ± 0.06 | 0.75 ±0.24 | **0.48±0.12** | 2.27±0.19 |
| | CNEE | 0.73 ± 0.09 | 1.12 ± 0.30 | **0.47±0.11** | 3.81±0.24 |
| BA Sim | PEHNE | 0.80 ± 0.08 | 0.66 ± 0.11 | **0.64±0.13** | 0.89±0.13 |
| | CNEE | 0.85 ± 0.08 | 0.81 ± 0.11 | **0.67±0.13** | 1.33±0.16 |
| Homophily Sim | PEHNE | 0.22 ± 0.02 | 0.34 ± 0.07 | **0.17±0.01** | 0.62±0.09 |
| | CNEE | 0.23 ± 0.02 | 0.35 ± 0.06 | **0.17±0.01** | 0.60±0.08 |

Table 5: Test set results (mean ± SD over five different initializations) for HINet with different GNN architectures. Lower is better for both metrics. The best-performing method is in bold; the second-best is underlined.

The results in Table 5 show that GCN often performs poorly compared to the other architectures. GraphSAGE achieves the best performance across all datasets, closely followed by GIN. However, this does not imply that GraphSAGE is intrinsically superior for treatment effect estimation in networks. As shown in Appendix G.3, using a different exposure mapping in the DGP yields a considerably different ranking, with GIN substantially outperforming the other architectures.

## 6 CONCLUSION

We introduced HINet, a novel method for estimating heterogeneous treatment effects in network settings with interference. In contrast to many prior works that rely on predefined exposure mappings, HINet learns directly from data how the treatments of an instance's neighbors influence its outcome. We empirically demonstrated the benefits of learning the exposure mapping and showed that a mismatch between the true exposure mapping (DGP) and an imposed one during modeling can have a profound impact—an issue resolved when the mapping is learned from data. Additionally, we introduced the notion of network-level covariate shift, which arises from the interaction between homophily and the treatment assignment mechanism, and we empirically showed that HINet's balanced node representations substantially mitigate this covariate shift's impact on estimation error.

**Limitations.** Representation balancing may introduce bias in treatment effect estimates if the learned representations are not invertible, as this can make the treatment effect non-identifiable (Melnychuk et al., 2023; 2025). Although HINet achieves state-of-the-art empirical performance, its representations are not guaranteed to be invertible, which may lead to biased treatment effect estimates. Future work could explore alternative approaches for estimating treatment effects under unknown exposure mappings that provide theoretical guarantees for the expected estimation error.

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

## A PROOF OF BALANCED REPRESENTATIONS WITH RESPECT TO TREATMENTS OF NEIGHBORS

**Proposition A.1.** *If the hidden representations $(\phi_i, \phi_{\mathcal{N}_i})$ are treatment-invariant with respect to $T_i$, i.e., $p(\phi_i, \phi_{\mathcal{N}_i} \mid T_i) = p(\phi_i, \phi_{\mathcal{N}_i})$, then these representations are also invariant with respect to the treatments of each of its neighbors: $p(\phi_i, \phi_{\mathcal{N}_i} \mid T_j) = p(\phi_i, \phi_{\mathcal{N}_i}) \quad \forall i \in \mathcal{V}, j \in \mathcal{N}_i$.*

*Proof.* For any neighbor $j$ of $i$ define $\mathcal{A}_{ij}$ and $\mathcal{B}_{ij}$:

$$\mathcal{A}_{ij} := \mathcal{N}_i \cap \mathcal{N}_j, \qquad \mathcal{B}_{ij} := \mathcal{N}_i \setminus \left(\mathcal{A}_{ij} \cup \{j\}\right),$$

so that $\mathcal{N}_i = \mathcal{A}_{ij} \cup \mathcal{B}_{ij} \cup \{j\}$.

We explicitly train our model to balance the representations for each node $i$ as follows:

$$(\phi_i, \phi_{\mathcal{N}_i}) \perp T_i, \quad \text{i.e.,} \quad p(\phi_i, \phi_{\mathcal{N}_i} \mid T_i) = p(\phi_i, \phi_{\mathcal{N}_i}) \quad \forall i \in \mathcal{V}. \tag{6}$$

Additionally, from the assumed causal graph and Markov assumption (see Section 3), i.e., only one-hop neighbors causally affect each other, we have that the features of the nodes in set $\mathcal{B}_{ij}$—which are neighbors of node $i$ that are not directly connected to node $j$—are independent of $T_j$:

$$\mathbf{X}_{\mathcal{B}_{ij}} \perp T_j.$$

Therefore, $T_j$ is also independent of the representations learned via $\phi_{\mathcal{B}_{ij}} = e_\phi(\mathbf{X}_{\mathcal{B}_{ij}})$:

$$\phi_{\mathcal{B}_{ij}} \perp T_j. \tag{7}$$

Write the collection $(\phi_i, \phi_{\mathcal{N}_i})$ as

$$(\phi_i, \phi_{\mathcal{N}_i}) \equiv (\phi_i, \phi_j, \phi_{\mathcal{A}_{ij}}, \phi_{\mathcal{B}_{ij}}).$$

Apply the chain rule to the joint distribution conditional on $T_j$:

$$p(\phi_i, \phi_j, \phi_{\mathcal{A}_{ij}}, \phi_{\mathcal{B}_{ij}} \mid T_j) = p(\phi_{\mathcal{B}_{ij}} \mid \phi_i, \phi_j, \phi_{\mathcal{A}_{ij}}, T_j)\, p(\phi_i, \phi_j, \phi_{\mathcal{A}_{ij}} \mid T_j). \tag{8}$$

By 7:

$$p(\phi_{\mathcal{B}_{ij}} \mid \phi_i, \phi_j, \phi_{\mathcal{A}_{ij}}, T_j) = p(\phi_{\mathcal{B}_{ij}} \mid \phi_i, \phi_j, \phi_{\mathcal{A}_{ij}}). \tag{9}$$

Next, since $(\phi_i, \phi_j, \phi_{\mathcal{A}_{ij}})$ is a subset of $(\phi_j, \phi_{\mathcal{N}_j})$, Equation (6) for node $j$ implies that

$$p(\phi_i, \phi_j, \phi_{\mathcal{A}_{ij}} \mid T_j) = p(\phi_i, \phi_j, \phi_{\mathcal{A}_{ij}}). \tag{10}$$

Substituting these two equalities into Equation (8) yields

$$p(\phi_i, \phi_j, \phi_{\mathcal{A}_{ij}}, \phi_{\mathcal{B}_{ij}} \mid T_j) = p(\phi_{\mathcal{B}_{ij}} \mid \phi_i, \phi_j, \phi_{\mathcal{A}_{ij}})\, p(\phi_i, \phi_j, \phi_{\mathcal{A}_{ij}}). \tag{11}$$

Now, applying the chain rule again, we have that

$$p(\phi_i, \phi_j, \phi_{\mathcal{A}_{ij}}, \phi_{\mathcal{B}_{ij}} \mid T_j) = p(\phi_i, \phi_j, \phi_{\mathcal{A}_{ij}}, \phi_{\mathcal{B}_{ij}}). \tag{12}$$

Therefore,

$$p(\phi_i, \phi_{\mathcal{N}_i} \mid T_j) = p(\phi_i, \phi_{\mathcal{N}_i}), \tag{13}$$

which is the claimed independence from $T_j$. $\qquad \square$

If there is homophily, the assumptions on the causal graph slightly change. Therefore, the proof has to be slightly modified as explained in Appendix B.2. Nevertheless, Proposition A.1 still holds.

## B HOMOPHILY AND INTERFERENCE

### B.1 CAUSAL DIAGRAM

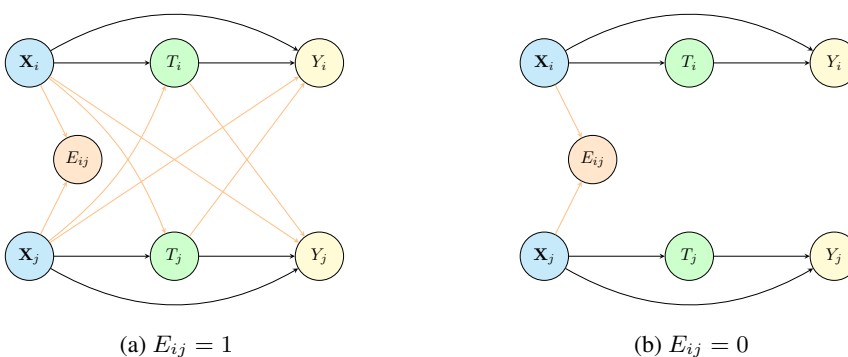

(a) $E_{ij} = 1$         (b) $E_{ij} = 0$

Figure 6: DAGs representing the causal structure when homophily is present. Conditioning on the presence of an edge reveals the underlying causal structure.

In Figure 6, we depict the causal DAGs that arise when homophily is present (Shalizi & Thomas, 2011). In contrast to Figure 2, the presence of an edge between nodes $i$ and $j$—denoted by the binary variable $E_{ij}$—is now itself a random variable that depends on the features of $i$ and $j$. When $E_{ij} = 1$, there can be direct causal (interference) effects from $i$ to $j$; when $E_{ij} = 0$, these interference effects are absent. Such DAGs are known as labeled, or context-specific DAGs (Pensar et al., 2015), meaning that the structure of the DAG depends on the value of a conditioned variable. In Figure 6, the two possible DAGs corresponding to the two values of $E_{ij}$ are shown. Importantly, conditioning on a node $i$'s one-hop neighborhood implicitly conditions on the edge variables in $\{E_{ik}\}_{k \in \mathcal{V}}$, which act as colliders, inducing an association between the features $\mathbf{X}_i$ and $\mathbf{X}_k$ of these nodes (Pearl, 2009). However, under the strong ignorability assumption,

$$Y_i(T_i = t_i, \mathbf{T}_{\mathcal{N}_i} = \mathbf{t}_{\mathcal{N}_i}) \perp\!\!\!\perp T_i, \mathbf{T}_{\mathcal{N}_i} \mid \mathbf{X}_i, \mathbf{X}_{\mathcal{N}_i}, \ \forall t_i \in \mathcal{T}, \mathbf{t}_{\mathcal{N}_i} \in \mathcal{T}^{\mathcal{N}_i}, \mathbf{X}_i \in \mathcal{X}, \mathbf{X}_{\mathcal{N}_i} \in \mathcal{X}^{\mathcal{N}_i},$$

this induced association does not jeopardize the identifiability of the treatment effects. Conditioning on $\mathbf{X}_i$ and $\mathbf{X}_{\mathcal{N}_i}$ still d-separates $Y_i$ and $T_j$ for all $i$ and $j$.

### B.2 BALANCED REPRESENTATIONS WITH RESPECT TO TREATMENTS OF NEIGHBORS

Proposition A.1 still holds when homophily is present, but its proof requires a small modification. Due to homophily, the independence $\mathbf{X}_{\mathcal{B}_{ij}} \perp T_j$ no longer holds after we condition on one-hop neighborhoods. Conditioning on the neighborhoods of $i$ and $j$, which we do in our proof, amounts to conditioning on all edges in $\{E_{ki}\}_{k \in \mathcal{V}} \cup \{E_{kj}\}_{k \in \mathcal{V}}$. More specifically, the problem is that we condition on the binary edge indicator variables $E_{bi}$ and $E_{bj}$ for every $b \in \mathcal{B}_{ij}$. These variables act as colliders that create a pathway from each $\mathbf{X}_b$ to $\mathbf{X}_i$ and $\mathbf{X}_j$. Concretely, conditioning on the neighborhoods of $i$ and $j$ opens the paths

$$\mathbf{X}_b \to E_{bi} \leftarrow \mathbf{X}_i \to T_j \qquad \text{and} \qquad \mathbf{X}_b \to E_{bj} \leftarrow \mathbf{X}_j \to T_j \quad \forall b \in \mathcal{B}_{ij},$$

which creates an association between $\mathbf{X}_{\mathcal{B}_{ij}}$ and $T_j$. Note that we do not condition on other edges than the ones in the set $\{E_{ki}\}_{k \in \mathcal{V}} \cup \{E_{kj}\}_{k \in \mathcal{V}}$, meaning no associations between $b$ and other nodes in the graph are induced. Additionally, given that by construction $E_{bj} = 0$ for all $b \in \mathcal{B}_{ij}$, there is no direct causal effect from $\mathbf{X}_{\mathcal{B}_{ij}}$ to $T_j$.

Now, to recover the conditional independence of $\mathbf{X}_{\mathcal{B}_{ij}}$ and $T_j$ needed for the proof, we must close these paths by conditioning on $\mathbf{X}_i$ and $\mathbf{X}_j$. This yields

$$\mathbf{X}_{\mathcal{B}_{ij}} \perp\!\!\!\perp T_j \mid \mathbf{X}_i, \mathbf{X}_j,$$

and hence

$$\phi_{\mathcal{B}_{ij}} \perp\!\!\!\perp T_j \mid \phi_i, \phi_j.$$

Using the above conditional independence gives:

$$p(\phi_{\mathcal{B}_{ij}} \mid \phi_i, \phi_j, \phi_{\mathcal{A}_{ij}}, T_j) = p(\phi_{\mathcal{B}_{ij}} \mid \phi_i, \phi_j, \phi_{\mathcal{A}_{ij}})$$

The remaining steps of the proof remain unchanged.

## C    PERFORMANCE METRICS

In a traditional no-interference setting with binary treatment, there is only one counterfactual: the outcome under the opposite treatment, $Y_i(1 - t_i)$. In network settings, however, counterfactuals must be considered at the level of the entire network, since the potential outcome of any given unit may depend on the treatments of others. Therefore, we argue that a good evaluation procedure should account for *counterfactual networks* rather than only individual-level counterfactuals. A counterfactual network is a network in which at least one unit receives a different treatment than in the observed network. Note that the number of counterfactual networks is $2^{|\mathcal{V}|} - 1$, each with $|\mathcal{V}|$ potential outcomes.

When simulated data is available, the Precision in Estimation of Heterogeneous Effects (PEHE) (Hill, 2011) is often used to evaluate methods in a traditional no-interference setting with binary treatment. PEHE is defined as the root mean squared error of the estimated Conditional Average Treatment Effects (CATEs), which is uniquely defined since there is a single counterfactual. In the presence of interference, however, this is no longer the case and estimated ITTEs could in principle be *evaluated for each counterfactual network* in a similar manner. For large networks, however, this becomes computationally intractable. Therefore, we propose two new metrics that sample a diverse set of counterfactual networks. These metrics are inspired by the Mean Integrated Squared Error (MISE), which is used for evaluating treatment effects with a continuous treatment (Schwab et al., 2020). In the continuous setting, a similar challenge arises due to the existence of more than two counterfactuals per unit.

The first proposed metric is the Precision in Estimation of Heterogeneous Network Effects (PEHNE), which evaluates ITTE estimation error over $m$ sampled counterfactual networks. The calculation of PEHNE is described in Algorithm 1.

---

**Algorithm 1** PEHNE calculation

---

1: **for** $j = 1, 2, ..., m$ **do**
2:     Percentage of nodes to treat $p_j = \frac{100 \cdot j}{m}\%$
3:     Sample treatment for each node $i$: $t_i^j \sim \text{Bernoulli}(p_j)$
4:     Estimate ITTE for each node $i$: $\hat{\omega}_i^j(t_i^j, \mathbf{t}_{\mathcal{N}_i}^j) = \hat{y}_i^j(t_i^j, \mathbf{t}_{\mathcal{N}_i}^j) - \hat{y}_i^j(0, \mathbf{0})$
5:     Calculate $\text{MSE}_j = \frac{1}{|\mathcal{V}|} \sum_i (\omega_i^j(t_i^j, \mathbf{t}_{\mathcal{N}_i}^j) - \hat{\omega}_i^j(t_i^j, \mathbf{t}_{\mathcal{N}_i}^j))^2$
6: **end for**
7: **return** PEHNE $= \frac{1}{m} \sum_j \text{MSE}_j$

---

The second proposed metric is the Counterfactual Network Estimation Error (CNEE), which evaluates counterfactual outcome estimation error over $m$ sampled counterfactual networks. The calculation of CNEE is described in Algorithm 2.

---

**Algorithm 2** CNEE calculation

---

1: **for** $j = 1, 2, ..., m$ **do**
2:     Percentage of nodes to treat $p_j = \frac{100 \cdot j}{m}\%$
3:     Sample treatment for each node $i$: $t_i^j \sim \text{Bernoulli}(p_j)$
4:     Estimate the potential outcome $Y_i(t_i, \mathbf{t}_{\mathcal{N}_i})$ for each node $i$: $\hat{y}_i^j(t_i^j, \mathbf{t}_{\mathcal{N}_i}^j)$
5:     Calculate $\text{MSE}_j = \frac{1}{|\mathcal{V}|} \sum_i (y_i^j(t_i^j, \mathbf{t}_{\mathcal{N}_i}^j) - \hat{y}_i^j(t_i^j, \mathbf{t}_{\mathcal{N}_i}^j))^2$
6: **end for**
7: **return** CNEE $= \frac{1}{m} \sum_j \text{MSE}_j$

---

By sampling treatments according to the percentage $p_j$, we ensure that models are evaluated across a variety of treatment rates. However, PEHNE places stronger emhasis on the estimation of the "zero" counterfactual network, i.e., the network in which no unit receives treatment, because of the term $Y_i(0, \mathbf{0})$ in Equation (1). If a model estimates these outcomes poorly, its performance in terms of PEHNE will be significantly penalized. In contrast, CNEE assigns equal importance to all sampled counterfactual networks.

In our experiments, we set $m = 50$ for both PEHNE and CNEE. Note that these metrics are used solely for performance evaluation, not for hyperparameter tuning (see Appendix F), as they cannot be calculated in practice from observational data—they require that all potential outcomes be known. Consequently, validation PEHNE/CNEE cannot be used for hyperparameter selection.

## D    DATA-GENERATING PROCESS

We adjust the DGP proposed by Jiang & Sun (2022) and Caljon et al. (2025). Instead of using a pre-defined exposure mapping $z_i = \frac{1}{|\mathcal{N}_i|} \sum_{j \in \mathcal{N}_i} t_j$, we define a function that allows for heterogeneous spillover effects.

For the fully synthetic datasets, we first generate $d$ features from a standard normal distribution: $x_i^j \sim \mathcal{N}(0, 1), j = 1, \ldots, d$. For the semi-synthetic datasets (Flickr and BC), we follow Jiang & Sun (2022) to partition each network into training, validation, and test sets using METIS (Karypis & Kumar, 1998). Then, following Guo et al. (2020); Jiang & Sun (2022), we use Latent Dirichlet Allocation (Blei et al., 2003) to reduce the sparse features to a lower-dimensional representation. Following the literature, we set the feature dimensionality to $d = 10$. More details on the network size of these two datasets are provided in Table 6.

|  | Flickr | | | BC | | |
|---|---|---|---|---|---|---|
|  | Train | Validation | Test | Train | Validation | Test |
| Nodes | 2,482 | 2,461 | 2,358 | 1,716 | 1,696 | 1,784 |
| Edges | 46,268 | 14,419 | 23,529 | 17,937 | 25,408 | 14,702 |

Table 6: Summary statistics for the Flickr and BC datasets.

For the fully synthetic datasets, we generate the network structure as follows. For the simulated BA dataset (BA Sim), each network of 5,000 nodes (i.e., training, validation, and test) is simulated based on the Barabási-Albert random network model (Barabási & Albert, 1999). The hyperparameter $m$ is set to 2. For the simulated homophilous dataset (Homophily Sim), homophilous networks with 5,000 nodes are generated based on the cosine similarity between the feature vectors of all node pairs in the network (some noise is added to the cosine similarity to allow unlikely edges to occur). Then, the node pairs are sorted according to cosine similarity. Edges are created between nodes with the highest cosine similarity until the average degree (number of edges per node) is equal to the average degree of the simulated BA network ($\bar{deg} = 4$).

To induce the causal structure (see Figure 2), we generate the following parameters:

$$
\begin{aligned}
w_j^{XT} &\sim \text{Unif}(-1, 1) \quad \text{for } j \in \{1, 2, \ldots, d\} & \mathbf{w}^{XT} &= [w_1^{XT}, w_2^{XT}, \ldots, w_d^{XT}], \\
w_j^{XY} &\sim \text{Unif}(-1, 1) \quad \text{for } j \in \{1, 2, \ldots, d\} & \mathbf{w}^{XY} &= [w_1^{XY}, w_2^{XY}, \ldots, w_d^{XY}], \\
w_j^{TY} &\sim \text{Unif}(-1, 1) \quad \text{for } j \in \{1, 2, \ldots, d\} & \mathbf{w}^{TY} &= [w_1^{TY}, w_2^{TY}, \ldots, w_d^{TY}], \\
w_j^{X_{\mathcal{N}}Y} &\sim \text{Unif}(-1, 1) \quad \text{for } j \in \{1, 2, \ldots, d\} & \mathbf{w}^{X_{\mathcal{N}}Y} &= [w_1^{X_{\mathcal{N}}Y}, w_2^{X_{\mathcal{N}}Y}, \ldots, w_d^{X_{\mathcal{N}}Y}], \\
w_j^{T_{\mathcal{N}}Y} &\sim \text{Unif}(-1, 1) \quad \text{for } j \in \{1, 2, \ldots, d\} & \mathbf{w}^{T_{\mathcal{N}}Y} &= [w_1^{T_{\mathcal{N}}Y}, w_2^{T_{\mathcal{N}}Y}, \ldots, w_d^{T_{\mathcal{N}}Y}].
\end{aligned}
$$

These parameters influence the effect of $\mathbf{X}_i$ on $T_i$, $\mathbf{X}_i$ on $Y_i$, the heterogeneous effect of $T_i$ on $Y_i$, the effect of $\mathbf{X}_{\mathcal{N}_i}$ on $Y_i$, and the heterogeneous spillover effect of $\mathbf{T}_{\mathcal{N}_i}$ on $Y_i$, respectively.

The treatment $t_i$ is generated as follows. We first calculate $\nu_i$ as:

$$
\nu_i = \beta_{XT} \cdot \mathbf{w}^{XT} \cdot \mathbf{x}_i,
$$

with $\beta_{XT} \geq 0$ the treatment assignment mechanism strength and $\mathbf{x}_i = [x_1, x_2, \ldots, x_d]'$. Next, to set the percentage of nodes treated to approximately 25%, we calculate the 75-th percentile $\nu_{75}$ and transform $\nu' = \nu - \nu_{75}$. Finally, we apply the sigmoid function $\sigma$ to $\nu'$, and obtain $t_i$ by sampling:

$$
t_i \sim \text{Bernoulli}(\sigma(\nu_i')).
$$

To generate the outcomes, we first create a transformed feature vector $\tilde{\mathbf{x}}_i$ by applying the sigmoid function $\sigma$ to half of the features in order to introduce nonlinearities. The outcomes are obtained as follows:

$$
y_i = \beta_{\text{individual}} \cdot h_i \cdot t_i + \beta_{\text{spillover}} \cdot z_i + \beta_{XY} \cdot u_i + \beta_{X_{\mathcal{N}}Y} \cdot u_{\mathcal{N}_i} + \beta_\epsilon \cdot \epsilon; \qquad \epsilon \sim \mathcal{N}(0, 1),
$$

with

$$h_i = \mathbf{w}^{TY} \cdot \tilde{\mathbf{x}}_i, \qquad\qquad u_i = \mathbf{w}^{XY} \cdot \tilde{\mathbf{x}}_i,$$

$$z_i = \frac{1}{|\mathcal{N}_i|} \sum_{j \in \mathcal{N}_i} t_j \cdot \mathbf{w}^{T_\mathcal{N} Y} \cdot \tilde{\mathbf{x}}_j, \qquad\qquad u_{\mathcal{N}_i} = \frac{1}{|\mathcal{N}_i|} \sum_{j \in \mathcal{N}_i} \mathbf{w}^{X_\mathcal{N} Y} \cdot \tilde{\mathbf{x}}_j.$$

Unless explicitly specified otherwise, we use the following parameter values in the experiments presented in Section 5: $\beta_{XT} = 6, \beta_{\text{individual}} = 2, \beta_{\text{spillover}} = 2, \beta_{XY} = 1.5, \beta_{X_\mathcal{N} Y} = 1.5, \beta_\epsilon = 0.2$.

## E  MEASURING HOMOPHILY

Homophily (McPherson et al., 2001), also known as assortative mixing (Newman, 2002; 2003), refers to the tendency of nodes in a network to associate with similar nodes. For example, individuals with similar interests are more likely to be friends. The degree of assortative mixing for a given feature can be quantified using the assortativity coefficient, which lies between -1 and 1. It is positive when an attribute is assortative, negative when it is disassortative, and zero when there is no assortativity. By calculating this coefficient for the treatment variable, we can objectively assess whether treated nodes are more likely to have treated neighbors. Similarly, outcome assortativity is positive when the outcomes of neighbors are positively correlated. We report these measures in Table 1 for the four datasets used in our experiments (Section 5).

To calculate assortativity for a categorical attribute, a mixing matrix, $M$, is defined, where the element $M_{ij}$ represents the fraction of all edges in the network that connect a node of category $i$ to a node of category $j$. The trace of this matrix, $\text{Tr}(M) = \sum_i M_{ii}$, thus quantifies the total fraction of edges connecting nodes within the same category. To assess whether this observed fraction is greater than what would be expected by chance, it is compared against the expected fraction of within-category edges of a network with random connections, which is given by $\sum_i a_i b_i$, where $a_i = \sum_j M_{ij}$ is the fraction of edges starting at nodes of category $i$, and $b_i = \sum_j M_{ji}$ is the fraction of edges ending at nodes of category $i$. For undirected networks, the matrix $M$ is symmetric, and therefore $a_i = b_i$. The assortativity coefficient, $r$, is then defined as the normalized difference between the observed and expected fractions of within-category edges (Newman, 2003):

$$r = \frac{\sum_i M_{ii} - \sum_i a_i b_i}{1 - \sum_i a_i b_i}. \tag{14}$$

A value of $r > 0$ indicates an assortative network, where connections occur more frequently between similar nodes. A value of $r < 0$ indicates a disassortative network, where connections occur more frequently between dissimilar nodes. A value of $r = 0$ signifies that connections are random with respect to the attribute.

For numerical attributes, the calculation of $r$ is slightly different, but the interpretation remains the same. For more details, we refer to Newman (2003).

## F  HYPERPARAMETER SELECTION AND IMPLEMENTATION DETAILS

Due to the fundamental problem of causal inference (Holland, 1986), individualized treatment effects are unobservable. As a result, selecting hyperparameters is challenging, since we cannot directly optimize based on treatment effect estimation error. For standard machine learning hyperparameters, such as hidden layer size or learning rate, we can rely on the factual validation loss for hyperparameter selection. The factual validation loss is the average estimation error for outcomes actually observed in the validation set and can always be calculated. However, the factual loss may not reflect the treatment effect estimation performance. Nevertheless, this approach has been shown to work reasonably well (Curth & van der Schaar, 2023).

The weight for adversarial balancing, $\alpha$, is a special type of hyperparameter. A positive $\alpha$ may cause the model to discard relevant information for predicting the observed outcomes in favor of constructing treatment-invariant representations, which will likely impair the factual validation loss. Consequently, if the factual loss is used to select this hyperparameter, $\alpha$ will often be chosen as zero—meaning that the upper branch of HINet would not be used. However, both theoretical and empirical work suggests that balancing representations can improve treatment effect estimates (Shalit et al., 2017; Bica et al., 2020; Berrevoets et al., 2020). Based on this, we propose the following approach for hyperparameter selection. First, the standard machine learning hyperparameters are tuned using the factual validation loss. Once these hyperparameters are set, the factual loss is calculated for different values of $\alpha$. As $\alpha$ increases, the factual loss typically increases as well. Our intuition is that a modest increase in factual loss is acceptable and merely indicates the representations have become more treatment-invariant. However, a substantial increase may suggest that valuable information is being discarded in favor of learning treatment-invariant representations. As a heuristic, we propose selecting the largest value of $\alpha$ for which the factual loss remains below $(1+p) \cdot \mathrm{loss}_{\alpha=0}$. As a rule of thumb, we set $p = 0.10$, meaning that we allow for a maximum increase in validation error of 10%. An important advantage of this approach is that it allows $\alpha = 0$ to be selected when representation balancing would otherwise result in excessive information being discarded.

For HINet, NetEst, and SPNet, the range for $\alpha$ is $\{0, 0.025, 0.05, 0.1, 0.2, 0.3\}$. The other hyperparameters are selected from the ranges shown in Table 7.

| Parameter | Value |
|---|---|
| Hidden size | $\{16, 32\}$ |
| Num. epochs | $\{500, 1000, 2000\}$ |
| Initial learning rate | $\{0.001, 0.0005, 0.0001\}$ |
| Dropout probability | $\{0.0, 0.1, 0.2\}$ |

Table 7: Hyperparameter ranges.

All GIN layers internally use a 2-layer MLP. The encoder block $e_\phi$ in HINet consists of two hidden layers, whereas the MLP blocks $d_T$ and $p_Y$, as well as the MLP block in the GIN model, each consist of three hidden layers. All MLP blocks (for every method) use ReLU activations after each layer. Other hyperparameters are set to author-recommended values. Each model is trained using the Adam optimizer (Kingma & Ba, 2015) with weight decay set to 0.001. For all models except TNet and SPNet, we use the implementation provided by Jiang & Sun (2022). Since there is no publicly available implementation of SPNet, we implemented it ourselves based on the description in Zhao et al. (2024). For TNet, the implementation from Chen et al. (2009) is used and $\alpha = \gamma = 1$ because it gave stable results in terms of factual validation loss for all datasets.

All reported results are averages over five different initializations, affecting both weight initialization and training data shuffling.

**Reproducibility.** Our code is available at https://anonymous.4open.science/r/HINet-12C5.

# G ADDITIONAL RESULTS

## G.1 IMPACT OF HOMOPHILY

In Figure 7, we visualize the impact of representation balancing on the test set results in terms of PEHNE. The results are similar to those presented in Section 5.2 for CNEE (Figure 4). When only individual (direct) effects are present, representation balancing improves performance for both the homophilous and non-homophilous networks with high values of $\beta_{XT}$. However, when spillover effects are present, differences between the non-homophilous and homophilous networks emerge. Specifically, when only spillover effects are present, balancing considerably improves performance for the homophilous networks but has little effect on the non-homophilous networks. When both individual and spillover effects are present, the performance gain from balancing is relatively larger under homophily.

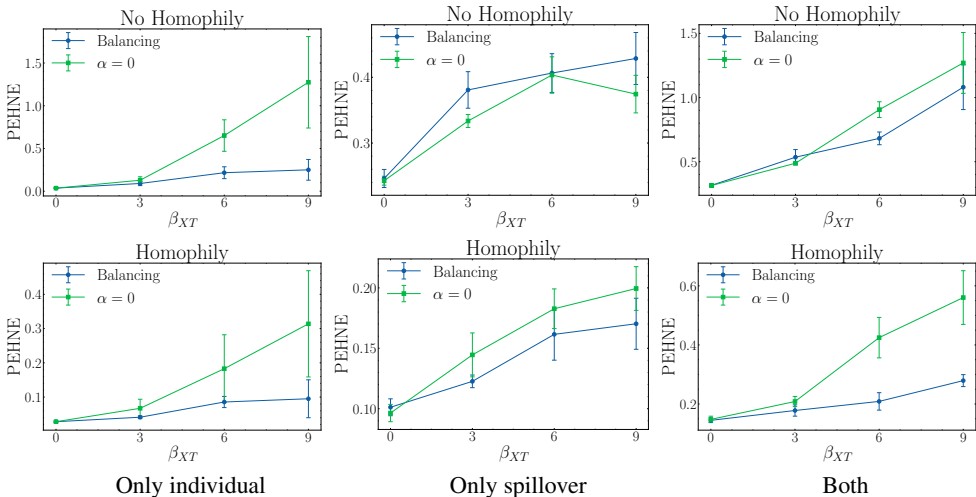

Figure 7: Impact of balancing node representations on test PEHNE (mean $\pm$ SD over five different initializations). The two rows correspond to the BA and Homophily Sim datasets. The columns correspond to different DGPs. The x-axis shows increasing treatment assignment mechanism strength $\beta_{XT}$. Lower values indicate better performance.

## G.2 LEARNING DIFFERENT EXPOSURE MAPPINGS

### G.2.1 SETTING THE INFLUENCE OF NEIGHBOR FEATURES TO ZERO

As discussed in Section 5.3, it is surprising that TNet and NetEst perform worse than HINet when their assumed exposure mapping is correctly specified. A likely reason is that they are unable to capture the nonlinear nature of the effect of $\mathbf{X}_{\mathcal{N}_i}$ on $Y_i$, which is present in the default DGP used in our experiments (see Appendix D). In the DGP, the strength of this effect is controlled by the parameter $\beta_{X_{\mathcal{N}}Y}$. To further analyze the impact of using the correctly specified exposure mapping on the estimation accuracy of TNet and NetEst, we set this parameter to zero. The results for this simplified DGP, shown in Table 8, indicate that TNet achieves the best performance in this setting.

| Dataset | Metric | TARNet | NetDeconf | NetEst | TNet | GIN model | SPNet | HINet (ours) |
|---|---|---|---|---|---|---|---|---|
| BC | PEHNE | $3.98 \pm 0.02$ | $5.31 \pm 0.11$ | $1.26 \pm 0.10$ | $\mathbf{0.54 \pm 0.09}$ | $\underline{0.97 \pm 0.09}$ | $5.80 \pm 0.65$ | $1.18 \pm 0.29$ |
|  | CNEE | $3.42 \pm 0.01$ | $4.66 \pm 0.08$ | $1.23 \pm 0.13$ | $\mathbf{0.56 \pm 0.11}$ | $\underline{0.90 \pm 0.10}$ | $5.06 \pm 0.38$ | $1.12 \pm 0.29$ |
| Flickr | PEHNE | $4.05 \pm 0.01$ | $4.07 \pm 0.06$ | $4.28 \pm 0.86$ | $\underline{0.90 \pm 0.53}$ | $\mathbf{0.85 \pm 0.11}$ | $5.19 \pm 0.24$ | $2.93 \pm 0.31$ |
|  | CNEE | $3.45 \pm 0.00$ | $3.83 \pm 0.08$ | $4.23 \pm 0.89$ | $\underline{0.89 \pm 0.56}$ | $\mathbf{0.82 \pm 0.11}$ | $5.30 \pm 0.26$ | $2.92 \pm 0.32$ |
| BA Sim | PEHNE | $3.51 \pm 0.01$ | $3.80 \pm 0.01$ | $1.15 \pm 0.51$ | $\mathbf{0.03 \pm 0.00}$ | $\underline{0.47 \pm 0.03}$ | $3.79 \pm 0.47$ | $1.38 \pm 0.19$ |
|  | CNEE | $2.92 \pm 0.01$ | $3.21 \pm 0.02$ | $1.15 \pm 0.51$ | $\mathbf{0.03 \pm 0.00}$ | $\underline{0.45 \pm 0.02}$ | $3.29 \pm 0.52$ | $1.40 \pm 0.20$ |
| Homophily Sim | PEHNE | $3.80 \pm 0.02$ | $1.53 \pm 0.06$ | $0.81 \pm 0.13$ | $\mathbf{0.03 \pm 0.01}$ | $\underline{0.36 \pm 0.09}$ | $3.66 \pm 0.32$ | $0.95 \pm 0.16$ |
|  | CNEE | $2.97 \pm 0.02$ | $1.20 \pm 0.05$ | $0.82 \pm 0.13$ | $\mathbf{0.04 \pm 0.01}$ | $\underline{0.36 \pm 0.09}$ | $3.39 \pm 0.30$ | $0.95 \pm 0.16$ |

Table 8: Test set results (mean $\pm$ SD over five different initializations) for the **proportion** of treated neighbors used as exposure mapping in the DGP, **and the influence of $\mathbf{X}_{\mathcal{N}_i}$ on $Y_i$ set to zero.** Lower is better for both metrics. The best-performing method is in bold; the second-best is underlined.

### G.2.2 ALTERNATIVE EXPOSURE MAPPINGS

To provide further evidence that HINet can learn a wide variety of different exposure mappings, we report results for two additional exposure mappings. In Table 9, we use information entropy as the exposure mapping: $z_i = -p \log_2(p) - (1-p) \log_2(1-p) - 0.5$, where $p$ is the proportion of treated neighbors. Additionally, we subtract 0.5 to allow for negative spillover effects. In Table 10, we use the squared weighted average exposure: $z_i = \frac{1}{|\mathcal{N}_i|} \sum_{j \in \mathcal{N}_i} w^2(\mathbf{x}_j) t_j$. The results for both mappings indicate that HINet achieves the best performance, followed by the GIN model, highlighting the importance of the representational power of GINs in learning exposure mappings.

| Dataset | Metric | TARNet | NetDeconf | NetEst | TNet | GIN model | SPNet | HINet (ours) |
|---|---|---|---|---|---|---|---|---|
| BC | PEHNE | $4.37 \pm 0.03$ | $6.03 \pm 0.19$ | $2.43 \pm 0.23$ | $1.73 \pm 0.36$ | $\underline{1.46 \pm 0.08}$ | $5.27 \pm 0.11$ | $\mathbf{0.92 \pm 0.21}$ |
|  | CNEE | $4.48 \pm 0.05$ | $6.21 \pm 0.18$ | $2.61 \pm 0.22$ | $1.72 \pm 0.37$ | $\underline{1.52 \pm 0.12}$ | $5.31 \pm 0.33$ | $\mathbf{0.93 \pm 0.23}$ |
| Flickr | PEHNE | $4.01 \pm 0.07$ | $5.01 \pm 0.11$ | $2.50 \pm 0.05$ | $1.23 \pm 0.24$ | $\underline{1.19 \pm 0.06}$ | $6.29 \pm 0.43$ | $\mathbf{0.76 \pm 0.16}$ |
|  | CNEE | $7.17 \pm 0.21$ | $7.74 \pm 0.15$ | $4.10 \pm 0.07$ | $1.57 \pm 0.14$ | $\underline{1.15 \pm 0.09}$ | $9.49 \pm 0.39$ | $\mathbf{0.90 \pm 0.18}$ |
| BA Sim | PEHNE | $3.16 \pm 0.01$ | $3.77 \pm 0.03$ | $0.60 \pm 0.10$ | $\underline{0.27 \pm 0.16}$ | $1.23 \pm 0.05$ | $4.36 \pm 0.35$ | $\mathbf{0.25 \pm 0.07}$ |
|  | CNEE | $5.49 \pm 0.02$ | $5.85 \pm 0.03$ | $1.28 \pm 0.11$ | $\underline{0.72 \pm 0.16}$ | $1.08 \pm 0.03$ | $6.18 \pm 0.29$ | $\mathbf{0.24 \pm 0.06}$ |
| Homophily Sim | PEHNE | $3.11 \pm 0.03$ | $1.94 \pm 0.05$ | $0.41 \pm 0.08$ | $\mathbf{0.10 \pm 0.01}$ | $0.83 \pm 0.16$ | $1.89 \pm 0.08$ | $\underline{0.13 \pm 0.04}$ |
|  | CNEE | $3.68 \pm 0.03$ | $1.76 \pm 0.05$ | $0.55 \pm 0.06$ | $\underline{0.24 \pm 0.01}$ | $0.92 \pm 0.15$ | $1.65 \pm 0.10$ | $\mathbf{0.14 \pm 0.04}$ |

Table 9: Test set results (mean $\pm$ SD over five different initializations) for the **information entropy** of treatments of neighbors used as exposure mapping in the DGP. Lower is better for both metrics. The best-performing method is in bold; the second-best is underlined.

| Dataset | Metric | TARNet | NetDeconf | NetEst | TNet | GIN model | SPNet | HINet (ours) |
|---|---|---|---|---|---|---|---|---|
| BC | PEHNE | $3.64 \pm 0.04$ | $5.51 \pm 0.24$ | $1.87 \pm 0.17$ | $2.37 \pm 0.37$ | $\underline{1.82 \pm 0.20}$ | $4.67 \pm 0.15$ | $\mathbf{0.88 \pm 0.08}$ |
|  | CNEE | $4.05 \pm 0.02$ | $5.81 \pm 0.24$ | $1.88 \pm 0.14$ | $2.77 \pm 0.38$ | $\underline{1.76 \pm 0.25}$ | $4.85 \pm 0.06$ | $\mathbf{0.89 \pm 0.09}$ |
| Flickr | PEHNE | $3.80 \pm 0.06$ | $4.98 \pm 0.09$ | $3.02 \pm 0.10$ | $1.88 \pm 0.20$ | $\underline{1.50 \pm 0.09}$ | $6.05 \pm 0.58$ | $\mathbf{0.81 \pm 0.08}$ |
|  | CNEE | $5.45 \pm 0.02$ | $6.75 \pm 0.10$ | $4.36 \pm 0.14$ | $2.34 \pm 0.11$ | $\underline{1.42 \pm 0.09}$ | $7.79 \pm 0.57$ | $\mathbf{0.90 \pm 0.09}$ |
| BA Sim | PEHNE | $3.61 \pm 0.03$ | $4.11 \pm 0.02$ | $1.49 \pm 0.18$ | $\mathbf{1.09 \pm 0.04}$ | $1.74 \pm 0.04$ | $4.63 \pm 0.25$ | $\underline{1.31 \pm 0.20}$ |
|  | CNEE | $5.22 \pm 0.01$ | $5.77 \pm 0.04$ | $2.24 \pm 0.19$ | $\underline{1.45 \pm 0.03}$ | $1.60 \pm 0.03$ | $6.10 \pm 0.26$ | $\mathbf{1.33 \pm 0.19}$ |
| Homophily Sim | PEHNE | $3.84 \pm 0.04$ | $1.93 \pm 0.04$ | $1.20 \pm 0.05$ | $44.27 \pm 34.39$ | $\underline{1.13 \pm 0.05}$ | $2.22 \pm 0.17$ | $\mathbf{0.69 \pm 0.17}$ |
|  | CNEE | $4.31 \pm 0.02$ | $1.86 \pm 0.05$ | $1.35 \pm 0.06$ | $48.59 \pm 37.50$ | $\underline{1.15 \pm 0.05}$ | $2.08 \pm 0.15$ | $\mathbf{0.66 \pm 0.15}$ |

Table 10: Test set results (mean $\pm$ SD over five different initializations) for the **squared weighted average** of treatments of neighbors used as exposure mapping in the DGP. Lower is better for both metrics. The best-performing method is in bold; the second-best is underlined.

### G.3 ABLATION STUDY

To further investigate the importance of the GNN architecture used in HINet, we repeat the experiment from Section 5.4 using the sum exposure mapping, $z_i = \sum_{j \in \mathcal{N}_i} t_j$, instead of the default weighted average exposure mapping (see Appendix D). In contrast to the results reported in Section 5.4, the results in Table 11 show that the GIN architecture performs best for the sum exposure mapping. Moreover, GraphSAGE—which performed best for the weighted average exposure mapping, now performs poorly.

| Dataset | Metric | GIN | GAT | GraphSAGE | GCN |
|---|---|---|---|---|---|
| BC | PEHNE | **25.22 $\pm$ 1.69** | 266.38 $\pm$17.04 | 392.30$\pm$14.5 | 69.63$\pm$7.43 |
|  | CNEE | **23.57 $\pm$ 1.28** | 256.37$\pm$ 13.65 | 357.89$\pm$16.95 | 60.23$\pm$6.49 |
| Flickr | PEHNE | **229.72 $\pm$37.89** | 2586.48 $\pm$154.24 | 2628.21$\pm$87.46 | 2582.09$\pm$145.21 |
|  | CNEE | **231.86 $\pm$ 32.76** | 2529.27$\pm$ 397.00 | 2384.61$\pm$89.27 | 1235.13$\pm$67.43 |
| BA Sim | PEHNE | **11.61 $\pm$ 0.69** | 54.68 $\pm$ 1.39 | 51.45$\pm$0.52 | 18.57$\pm$1.18 |
|  | CNEE | 12.20 $\pm$ 0.86 | 52.81 $\pm$ 0.74 | 49.12$\pm$0.74 | **10.69$\pm$0.52** |
| Homophily Sim | PEHNE | **0.20 $\pm$ 0.07** | 7.60 $\pm$ 0.16 | 7.77$\pm$0.21 | 3.10$\pm$0.20 |
|  | CNEE | **0.22 $\pm$ 0.07** | 7.48 $\pm$ 0.10 | 7.49$\pm$0.24 | 3.17$\pm$0.17 |

Table 11: Test set results (mean $\pm$ SD over five different initializations) for HINet with different GNN architectures, with the **sum** of treatments of neighbors used as exposure mapping in the DGP. Lower is better for both metrics. The best-performing method is in bold; the second-best is underlined.

## H LLM USAGE

In this paper, we used LLMs to polish the writing and as an assistant to write the code.

