# OpenReview forum: "Estimating Treatment Effects in Networks using Domain Adversarial Training"
_ICLR.cc/2026/Conference — ICLR 2026 Conference Withdrawn Submission_

### Official Review · Reviewer_GWF8 · 2025-10-21

**Soundness:** 2
**Presentation:** 3
**Contribution:** 2
**Rating:** 4
**Confidence:** 4

**Summary:**

This paper introduces HINet (Heterogeneous Interference Network), a novel method for estimating treatment effects in network settings where the outcome of one unit can be influenced by the treatment status of neighboring units (interference). The paper demonstrates HINet's effectiveness through extensive experiments on synthetic and semi-synthetic datasets, showing superior performance compared to existing methods across various exposure mapping scenarios.

**Strengths:**

The experimental design is comprehensive and well-executed, with the authors testing multiple exposure mapping functions including weighted averages, sums, entropy-based measures, and squared weighted averages. The introduction of two new evaluation metrics, PEHNE and CNEE, addresses a genuine gap in the literature regarding how to evaluate treatment effect estimators under interference across multiple counterfactual networks rather than just a single counterfactual scenario.

**Weaknesses:**

The most significant theoretical limitation is the lack of guarantees regarding the invertibility of learned representations. As the authors acknowledge, non-invertible representations can lead to biased treatment effect estimates, yet HINet provides no mechanism to ensure or verify invertibility. This is particularly concerning given that domain adversarial training explicitly removes information to achieve balance, potentially making the treatment effect non-identifiable.

**Questions:**

Can the authors provide any theoretical analysis or empirical diagnostics for assessing when the learned representations maintain sufficient information for treatment effect identifiability?

---

### Official Review · Reviewer_bv9d · 2025-10-28

**Soundness:** 1
**Presentation:** 1
**Contribution:** 3
**Rating:** 2
**Confidence:** 4

**Summary:**

The paper proposes HINet, a framework for estimating heterogeneous treatment effects in network settings where interference between units violates the traditional no-interference assumption. The method integrates Graph Neural Networks (GNNs) with domain adversarial training to learn both direct and spillover effects without relying on a predefined exposure mapping. It also introduces the concept of network-level covariate shift, which arises from the interaction between homophily and treatment assignment mechanisms, and states that this shift can bias treatment effect estimates. HINet addresses this challenge by learning balanced, treatment-invariant node representations that mitigate such bias. The authors present experiments on two semi-synthetic datasets, demonstrating that HINet achieves more accurate treatment effect estimation across network structures compared to existing approaches. The study also compares the performances in networks with and without homophily, and show that the proposed method has more advantage in homophilous networks.

**Strengths:**

- The paper addresses an important concern on network homophily in causal inference with interference. The introduction of the concept of network-level covariate shift due to homophily and treatment assignment interaction is a novel perspective.
- The proposed HINet framework is methodologically reasonable, combining Graph Neural Networks with counterfactual representation learning to jointly learn exposure effects and estimate treatment effects.
- Theoretical analysis and extensive empirical evaluation on semi-synthetic datasets provide evidence of the method’s effectiveness and robustness.

**Weaknesses:**

- The paper lacks rigorous formulation of several key concepts. For instance, the adversarial training framework is not clearly defined or formalized, and the notion of covariate shift in networks is not explicitly characterized. The relationship between network homophily and covariate shift also requires further mathematical clarification to support the proposed intuition.

- The algorithmic presentation omits some important implementation details. In particular, the paper does not clearly describe how representation balancing is adapted to the network structure, how the Graph Isomorphism Network (GIN) contributes to maximizing representational capacity, or how the model is trained and optimized in practice.

- The paper provides no theoretical analysis to support the empirical findings. Given that the method builds on counterfactual representation learning, it would be valuable to theoretically justify how balanced representations improve generalization in networked settings. Furthermore, since the exposure mapping is assumed to exist and is learned from data, the paper should discuss whether the mapping can be consistently or accurately identified, and how misspecification might introduce bias in treatment effect estimation.

**Questions:**

### Extra Questions on Experiments

- In the ablation study, removing the balancing component (setting $\alpha = 0$) occasionally results in slightly worse performance, but not consistently across datasets. Could the authors elaborate on why representation balancing has only a limited or even negative effect in smaller datasets such as Flickr?

- The experiments suggest that homophily amplifies network-level covariate shift. Could the authors provide additional intuition or visual evidence illustrating how homophilous connections influence estimation error in practice?

- It would also be informative to assess how HINet performs under different network structures. Specifically, how does the model behave when the graph becomes extremely sparse or dense? Does performance degrade in either extreme, and if so, what are the underlying causes?

**Details Of Ethics Concerns:**

None.

---

### Official Review · Reviewer_QXac · 2025-10-31

**Soundness:** 2
**Presentation:** 3
**Contribution:** 1
**Rating:** 2
**Confidence:** 3

**Summary:**

The paper addresses the problem of unknown interference mechanisms in networks, where homophily and treatment assignment jointly induce network-level covariate shift.
To tackle this, the authors propose HINet, which integrates GIN-based neighborhood aggregation with domain-adversarial representation balancing, and validate its effectiveness on (semi-)synthetic network datasets.

**Strengths:**

The logical structure of the paper is clear,
and the exploration of homophily under network interference is significant.

**Weaknesses:**

1. The assumptions in Section 3 still rely on one-hop interference (Markov assumption) and strong ignorability.
If higher-order interference exists in real networks, the current Figure 2 and Equation (5) cannot adequately model those dependencies.
2. All results are based on semi-synthetic or fully synthetic datasets; adding a small robustness study on a real-world social graph with unobserved homophily would make the work more complete.
3. As shown in Sections 3 and 5.2, the paper only provides empirical evidence, no formal derivation of error bounds or theorem-level analysis for network-level covariate shift. So the support remains empirical.
4. While the appendix specifies 𝑚=50 counterfactual networks used solely for evaluation, there is no justification for the sufficiency or convergence of this choice.
5. To ensure fairness, the authors could further clarify the hyperparameter search space and final settings for TNet and SPNet implementations.

**Questions:**

Please refer to the weaknesses section.

---

### Official Review · Reviewer_hAvC · 2025-10-31

**Soundness:** 3
**Presentation:** 2
**Contribution:** 2
**Rating:** 4
**Confidence:** 4

**Summary:**

This paper proposes an exposure mapping learning method for causal effect estimation with networked interference. The Graph Isomorphism Network and adversarial training are employed, and experiments on benchmark datasets are conducted to evaluate the performance of the proposed method.

**Strengths:**

The proposed method is generally reasonable.

**Weaknesses:**

1. The technical contributions are not significant. Both GIN and adversarial training are classical existing approaches. GIN has been widely used in graph learning, and adversarial learning is a standard method for distribution alignment. Exposure mapping learning has also been studied in the literature, as discussed in Section 2. What is the major difference and advantage of the submission compared with existing methods?

2. The key point of the submission is exposure mapping learning. What is the advantage of GIN and adversarial learning to exposure mapping learning? More discussions regarding this could improve the motivation.

3. The results in Tables 5 and 11 are questionable. GCN usually achieves promising performance in node classification. However, the GCN-based method does not obtain good performance in Tables 5 and 11. It would be better to discuss this more.

4. In addition, compared with Tables 5 and 11, what is the reason that the methods have quite different performance rankings with different exposure mapping functions?

5. It would be better to analyze the statistical properties of the proposed method, such as the convergence, consistency, or unbiasedness, which are helpful to verify the advantage from a theoretical perspective.

6. The title of the submission is too normal and fails to highlight the key point. It seems that the core idea is exposure mapping learning rather than adversarial training.

7. Some notations in Figure 3 have not been described.

**Questions:**

Please refer to the weaknesses.

---

### Author Response · Authors · 2025-11-19

We want to thank the reviewers for their comments and feedback on our manuscript. Those have been valuable pointers on how to improve and update our work to enhance clarity, contribution, and impact.

Based on the reviews, we have decided to withdraw our paper.

---

### Note · Authors · 2025-11-19

I have read and agree with the venue's withdrawal policy on behalf of myself and my co-authors.